# Gaussian Process Random Fields

**David A. Moore and Stuart J. Russell**
Computer Science Division
University of California, Berkeley
Berkeley, CA 94709
{dmoore, russell}@cs.berkeley.edu

## Abstract

Gaussian processes have been successful in both supervised and unsupervised machine learning tasks, but their computational complexity has constrained practical applications. We introduce a new approximation for large-scale Gaussian processes, the Gaussian Process Random Field (GPRF), in which local GPs are coupled via pairwise potentials. The GPRF likelihood is a simple, tractable, and parallelizeable approximation to the full GP marginal likelihood, enabling latent variable modeling and hyperparameter selection on large datasets. We demonstrate its effectiveness on synthetic spatial data as well as a real-world application to seismic event location.

## 1 Introduction

Many machine learning tasks can be framed as learning a function given noisy information about its inputs and outputs. In regression and classification, we are given inputs and asked to predict the outputs; by contrast, in latent variable modeling we are given a set of outputs and asked to reconstruct the inputs that could have produced them. Gaussian processes (GPs) are a flexible class of probability distributions on functions that allow us to approach function-learning problems from an appealingly principled and clean Bayesian perspective. Unfortunately, the time complexity of exact GP inference is $O(n^3)$, where $n$ is the number of data points. This makes exact GP calculations infeasible for real-world data sets with $n > 10000$.

Many approximations have been proposed to escape this limitation. One particularly simple approximation is to partition the input space into smaller blocks, replacing a single large GP with a multitude of local ones. This gains tractability at the price of a potentially severe independence assumption.

In this paper we relax the strong independence assumptions of independent local GPs, proposing instead a Markov random field (MRF) of local GPs, which we call a Gaussian Process Random Field (GPRF). A GPRF couples local models via pairwise potentials that incorporate covariance information. This yields a surrogate for the full GP marginal likelihood that is simple to implement and can be tractably evaluated and optimized on large datasets, while still enforcing a smooth covariance structure. The task of approximating the marginal likelihood is motivated by unsupervised applications such as the GP latent variable model [1], but examining the predictions made by our model also yields a novel interpretation of the Bayesian Committee Machine [2].

We begin by reviewing GPs and MRFs, and some existing approximation methods for large-scale GPs. In Section 3 we present the GPRF objective and examine its properties as an approximation to the full GP marginal likelihood. We then evaluate it on synthetic data as well as an application to seismic event location.

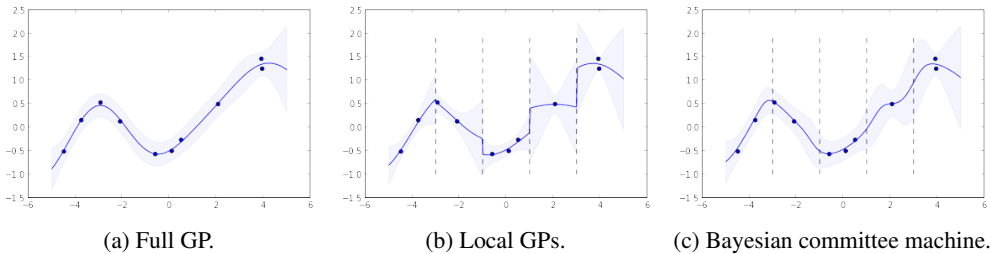

|   |   |   |
|:-:|:-:|:-:|
| (a) Full GP. | (b) Local GPs. | (c) Bayesian committee machine. |

Figure 1: Predictive distributions on a toy regression problem.

## 2 Background

### 2.1 Gaussian processes

Gaussian processes [3] are distributions on real-valued functions. GPs are parameterized by a mean function $\mu_\theta(\mathbf{x})$, typically assumed without loss of generality to be $\mu(\mathbf{x}) = 0$, and a covariance function (sometimes called a kernel) $k_\theta(\mathbf{x}, \mathbf{x}')$, with hyperparameters $\theta$. A common choice is the squared exponential covariance, $k_{SE}(\mathbf{x}, \mathbf{x}') = \sigma_f^2 \exp\left(-\frac{1}{2}\|\mathbf{x} - \mathbf{x}'\|^2/\ell^2\right)$, with hyperparameters $\sigma_f^2$ and $\ell$ specifying respectively a prior variance and correlation lengthscale.

We say that a random function $f(x)$ is Gaussian process distributed if, for any $n$ input points $X$, the vector of function values $\mathbf{f} = f(X)$ is multivariate Gaussian, $\mathbf{f} \sim \mathcal{N}(\mathbf{0}, k_\theta(X, X))$. In many applications we have access only to noisy observations $\mathbf{y} = \mathbf{f} + \varepsilon$ for some noise process $\varepsilon$. If the noise is iid Gaussian, i.e., $\varepsilon \sim \mathcal{N}(\mathbf{0}, \sigma_n^2 \mathcal{I})$, then the observations are themselves Gaussian, $\mathbf{y} \sim \mathcal{N}(\mathbf{0}, K_y)$, where $K_y = k_\theta(X, X) + \sigma_n^2 \mathcal{I}$.

The most common application of GPs is to Bayesian regression [3], in which we attempt to predict the function values $\mathbf{f}^*$ at test points $X^*$ via the conditional distribution given the training data, $p(\mathbf{f}^*|\mathbf{y}; X, X^*, \theta)$. Sometimes, however, we do not observe the training inputs $X$, or we observe them only partially or noisily. This setting is known as the Gaussian Process Latent Variable Model (GP-LVM) [1]; it uses GPs as a model for unsupervised learning and nonlinear dimensionality reduction. The GP-LVM setting typically involves multi-dimensional observations, $Y = (\mathbf{y}^{(1)}, \ldots, \mathbf{y}^{(D)})$, with each output dimension $\mathbf{y}^{(d)}$ modeled as an independent Gaussian process. The input locations and/or hyperparameters are typically sought via maximization of the *marginal likelihood*

$$\mathcal{L}(X, \theta) = \log p(Y; X, \theta) = \sum_{i=1}^{D} -\frac{1}{2}\log|K_y| - \frac{1}{2}\mathbf{y}_i^T K_y^{-1}\mathbf{y}_i + C$$

$$= -\frac{D}{2}\log|K_y| - \frac{1}{2}\text{tr}(K_y^{-1}YY^T) + C, \tag{1}$$

though some recent work [4, 5] attempts to recover an approximate posterior on $X$ by maximizing a variational bound. Given a differentiable covariance function, this maximization is typically performed by gradient-based methods, although local maxima can be a significant concern as the marginal likelihood is generally non-convex.

### 2.2 Scalability and approximate inference

The main computational difficulty in GP methods is the need to invert or factor the kernel matrix $K_y$, which requires time cubic in $n$. In GP-LVM inference this must be done at every optimization step to evaluate (1) and its derivatives.

This complexity has inspired a number of approximations. The most commonly studied are *inducing-point* methods, in which the unknown function is represented by its values at a set of $m$ inducing points, where $m \ll n$. These points can be chosen by maximizing the marginal likelihood in a surrogate model [6, 7] or by minimizing the KL divergence between the approximate and exact GP posteriors [8]. Inference in such models can typically be done in $O(nm^2)$ time, but this comes at the price of reduced representational capacity: while smooth functions with long lengthscales may be compactly represented by a small number of inducing points, for quickly-varying functions with

significant local structure it may be difficult to find any faithful representation more compact than the complete set of training observations.

A separate class of approximations, so-called "local" GP methods [3, 9, 10], involves partitioning the inputs into blocks of $m$ points each, then modeling each block with an independent Gaussian process. If the partition is spatially local, this corresponds to a covariance function that imposes independence between function values in different regions of the input space. Computationally, each block requires only $O(m^3)$ time; the total time is linear in the number of blocks. Local approximations preserve short-lengthscale structure within each block, but their harsh independence assumptions can lead to predictive discontinuities and inaccurate uncertainties (Figure 1b). These assumptions are problematic for GP-LVM inference because the marginal likelihood becomes discontinuous at block boundaries. Nonetheless, local GPs sometimes work very well in practice, achieving results comparable to more sophisticated methods in a fraction of the time [11].

The Bayesian Committee Machine (BCM) [2] attempts to improve on independent local GPs by averaging the predictions of multiple GP experts. The model is formally equivalent to an inducing-point model in which the *test points* are the inducing points, i.e., it assumes that the training blocks are conditionally independent given the test data. The BCM can yield high-quality predictions that avoid the pitfalls of local GPs (Figure 1c), while maintaining scalability to very large datasets [12]. However, as a purely predictive approximation, it is unhelpful in the GP-LVM setting, where we are interested in the likelihood of our training set irrespective of any particular test data. The desire for a BCM-style approximation to the marginal likelihood was part of the motivation for this current work; in Section 3.2 we show that the GPRF proposed in this paper can be viewed as such a model.

Mixture-of-experts models [13, 14] extend the local GP concept in a different direction: instead of deterministically assigning points to GP models based on their spatial locations, they treat the assignments as unobserved random variables and do inference over them. This allows the model to adapt to different functional characteristics in different regions of the space, at the price of a more difficult inference task. We are not aware of mixture-of-experts models being applied in the GP-LVM setting, though this should in principle be possible.

Simple building blocks are often combined to create more complex approximations. The PIC approximation [15] blends a global inducing-point model with local block-diagonal covariances, thus capturing a mix of global and local structure, though with the same boundary discontinuities as in "vanilla" local GPs. A related approach is the use of covariance functions with compact support [16] to capture local variation in concert with global inducing points. [11] surveys and compares several approximate GP regression methods on synthetic and real-world datasets.

Finally, we note here the similar title of [17], which is in fact orthogonal to the present work: they use a random field as a *prior* on input locations, whereas this paper defines a random field decomposition of the GP model itself, which may be combined with any prior on $X$.

### 2.3 Markov Random Fields

We recall some basic theory regarding Markov random fields (MRFs), also known as undirected graphical models [18]. A pairwise MRF consists of an undirected graph $(V, E)$, along with *node potentials* $\psi_i$ and *edge potentials* $\psi_{ij}$, which define an *energy function* on a random vector $\mathbf{y}$,

$$E(\mathbf{y}) = \sum_{i \in V} \psi_i(\mathbf{y}_i) + \sum_{(i,j) \in E} \psi_{ij}(\mathbf{y}_i, \mathbf{y}_j), \tag{2}$$

where $\mathbf{y}$ is partitioned into components $\mathbf{y}_i$ identified with nodes in the graph. This energy in turn defines a probability density, the "Gibbs distribution", given by $p(\mathbf{y}) = \frac{1}{Z}\exp(-E(\mathbf{y}))$ where $Z = \int \exp(-E(\mathbf{z}))d\mathbf{z}$ is a normalizing constant.

Gaussian random fields are the special case of pairwise MRFs in which the Gibbs distribution is multivariate Gaussian. Given a partition of $\mathbf{y}$ into sub-vectors $\mathbf{y}_1, \mathbf{y}_2, \ldots, \mathbf{y}_M$, a zero-mean Gaussian distribution with covariance $K$ and precision matrix $J = K^{-1}$ can be expressed by potentials

$$\psi_i(\mathbf{y}_i) = -\frac{1}{2}\mathbf{y}_i^T J_{ii}\mathbf{y}_i, \qquad \psi_{ij}(\mathbf{y}_i, \mathbf{y}_j) = -\mathbf{y}_i^T J_{ij}\mathbf{y}_j \tag{3}$$

where $J_{ij}$ is the submatrix of $J$ corresponding to the sub-vectors $\mathbf{y}_i, \mathbf{y}_j$. The normalizing constant $Z = (2\pi)^{n/2}|K|^{1/2}$ involves the determinant of the covariance matrix. Since edges whose potentials

are zero can be dropped without effect, the nonzero entries of the precision matrix can be seen as specifying the edges present in the graph.

## 3 Gaussian Process Random Fields

We consider a vector of $n$ real-valued[1] observations $\mathbf{y} \sim \mathcal{N}(\mathbf{0}, K_y)$ modeled by a GP, where $K_y$ is implicitly a function of input locations $X$ and hyperparameters $\theta$. Unless otherwise specified, all probabilities $p(\mathbf{y}_i), p(\mathbf{y}_i, \mathbf{y}_j)$, etc., refer to marginals of this full GP. We would like to perform gradient-based optimization on the marginal likelihood (1) with respect to $X$ and/or $\theta$, but suppose that the cost of doing so directly is prohibitive.

In order to proceed, we assume a partition $\mathbf{y} = (\mathbf{y}_1, \mathbf{y}_2, \dots, \mathbf{y}_M)$ of the observations into $M$ blocks of size at most $m$, with an implied corresponding partition $X = (X_1, X_2, \dots, X_M)$ of the (perhaps unobserved) inputs. The source of this partition is not a focus of the current work; we might imagine that the blocks correspond to spatially local clusters of input points, assuming that we have noisy observations of the $X$ values or at least a reasonable guess at an initialization. We let $K_{ij} = \mathrm{cov}_\theta(\mathbf{y}_i, \mathbf{y}_j)$ denote the appropriate submatrix of $K_y$, and $J_{ij}$ denote the corresponding submatrix of the precision matrix $J_y = K_y^{-1}$; note that $J_{ij} \neq (K_{ij})^{-1}$ in general.

### 3.1 The GPRF Objective

Given the precision matrix $J_y$, we could use (3) to represent the full GP distribution in factored form as an MRF. This is not directly useful, since computing $J_y$ requires cubic time. Instead we propose approximating the marginal likelihood via a random field in which local GPs are connected by pairwise potentials. Given an edge set which we will initially take to be the complete graph, $E = \{(i,j)|1 \le i < j \le M\}$, our approximate objective is

$$q_{GPRF}(\mathbf{y}; X, \theta) = \prod_{i=1}^{M} p(\mathbf{y}_i) \prod_{(i,j) \in E} \frac{p(\mathbf{y}_i, \mathbf{y}_j)}{p(\mathbf{y}_i)p(\mathbf{y}_j)}, \qquad (4)$$

$$= \prod_{i=1}^{M} p(\mathbf{y}_i)^{1-|E_i|} \prod_{(i,j) \in E} p(\mathbf{y}_i, \mathbf{y}_j)$$

where $E_i$ denotes the neighbors of $i$ in the graph, and $p(\mathbf{y}_i)$ and $p(\mathbf{y}_i, \mathbf{y}_j)$ are marginal probabilities under the full GP; equivalently they are the likelihoods of local GPs defined on the points $X_i$ and $X_i \cup X_j$ respectively. Note that these local likelihoods depend implicitly on $X$ and $\theta$. Taking the log, we obtain the energy function of an unnormalized MRF

$$\log q_{GPRF}(\mathbf{y}; X, \theta) = \sum_{i=1}^{M} (1 - |E_i|) \log p(\mathbf{y}_i) + \sum_{(i,j) \in E} \log p(\mathbf{y}_i, \mathbf{y}_j) \qquad (5)$$

with potentials

$$\psi_i^{GPRF}(\mathbf{y}_i) = (1 - |E_i|) \log p(\mathbf{y}_i), \qquad \psi_{ij}^{GPRF}(\mathbf{y}_i, \mathbf{y}_j) = \log p(\mathbf{y}_i, \mathbf{y}_j). \qquad (6)$$

We refer to the approximate objective (5) as $q_{GPRF}$ rather than $p_{GPRF}$ to emphasize that it is not in general a normalized probability density. It can be interpreted as a "Bethe-type" approximation [19], in which a joint density is approximated via overlapping pairwise marginals. In the special case that the full precision matrix $J_y$ induces a tree structure on the blocks of our partition, $q_{GPRF}$ recovers the exact marginal likelihood. (This is shown in the supplementary material.) In general this will not be the case, but in the spirit of loopy belief propagation [20], we consider the tree-structured case as an approximation for the general setting.

Before further analyzing the nature of the approximation, we first observe that as a sum of local Gaussian log-densities, the objective (5) is straightforward to implement and fast to evaluate. Each of the $O(M^2)$ pairwise densities requires $O((2m)^3) = O(m^3)$ time, for an overall complexity of

$O(M^2m^3) = O(n^2m)$ when $M = n/m$. The quadratic dependence on $n$ cannot be avoided by any algorithm that computes similarities between all pairs of training points; however, in practice we will consider "local" modifications in which $E$ is something smaller than all pairs of blocks. For example, if each block is connected only to a fixed number of spatial neighbors, the complexity reduces to $O(nm^2)$, i.e., linear in $n$. In the special case where $E$ is the empty set, we recover the exact likelihood of independent local GPs.

It is also straightforward to obtain the gradient of (5) with respect to hyperparameters $\theta$ and inputs $X$, by summing the gradients of the local densities. The likelihood and gradient for each term in the sum can be evaluated independently using only local subsets of the training data, enabling a simple parallel implementation.

Having seen that $q_{GPRF}$ can be optimized efficiently, it remains for us to argue its validity as a proxy for the full GP marginal likelihood. Due to space constraints we defer proofs to the supplementary material, though our results are not difficult. We first show that, like the full marginal likelihood (1), $q_{GPRF}$ has the form of a Gaussian distribution, but with a different precision matrix.

**Theorem 1.** *The objective $q_{GPRF}$ has the form of an unnormalized Gaussian density with precision matrix $\tilde{J}$, with blocks $\tilde{J}_{ij}$ given by*

$$\tilde{J}_{ii} = K_{ii}^{-1} + \sum_{j \in E_i} \left( Q_{11}^{(ij)} - K_{ii}^{-1} \right), \qquad \tilde{J}_{ij} = \left\{ \begin{array}{ll} Q_{12}^{(ij)} & if\ (i,j) \in E \\ 0 & otherwise. \end{array} \right), \qquad (7)$$

*where $Q^{(ij)}$ is the* local precision matrix $Q^{(ij)}$ *defined as the inverse of the marginal covariance,*

$$Q^{(ij)} = \left( \begin{array}{cc} Q_{11}^{(ij)} & Q_{12}^{(ij)} \\ Q_{21}^{(ij)} & Q_{22}^{(ij)} \end{array} \right) = \left( \begin{array}{cc} K_{ii} & K_{ij} \\ K_{ji} & K_{jj} \end{array} \right)^{-1}.$$

Although the Gaussian density represented by $q_{GPRF}$ is not in general normalized, we show that it is *approximately* normalized in a certain sense.

**Theorem 2.** *The objective $q_{GPRF}$ is approximately normalized in the sense that the optimal value of the* Bethe free energy *[19],*

$$F_B(b) = \sum_{i \in V} \left( \int_{\mathbf{y}_i} b_i(\mathbf{y}_i) \frac{(1 - |E_i|) \ln b_i(\mathbf{y}_i)}{\ln \psi_i(\mathbf{y}_i)} \right) + \sum_{(i,j) \in E} \left( \int_{\mathbf{y}_i, \mathbf{y}_j} b_{ij}(\mathbf{y}_i, \mathbf{y}_j) \ln \frac{b_{ij}(\mathbf{y}_i, \mathbf{y}_j)}{\psi_{ij}(\mathbf{y}_i, \mathbf{y}_j))} \right) \approx \log Z,$$
$$(8)$$

*the approximation to the normalizing constant found by loopy belief propagation, is precisely zero. Furthermore, this optimum is obtained when the pseudomarginals $b_i, b_{ij}$ are taken to be the true GP marginals $p_i, p_{ij}$.*

This implies that loopy belief propagation run on a GPRF would recover the marginals of the true GP.

### 3.2 Predictive equivalence to the BCM

We have introduced $q_{GPRF}$ as a surrogate model for the training set $(X, \mathbf{y})$; however, it is natural to extend the GPRF to make predictions at a set of test points $X^*$, by including the function values $\mathbf{f}^* = f(X^*)$ as an $M + 1$st block, with an edge to each of the training blocks. The resulting predictive distribution,

$$p_{GPRF}(\mathbf{f}^* | \mathbf{y}) \propto q_{GPRF}(\mathbf{f}^*, \mathbf{y}) = p(\mathbf{f}^*) \prod_{i=1}^{M} \frac{p(\mathbf{y}_i, \mathbf{f}^*)}{p(\mathbf{y}_i) p(\mathbf{f}^*)} \left( \prod_{i=1}^{M} p(\mathbf{y}_i) \prod_{(i,j) \in E} \frac{p(\mathbf{y}_i, \mathbf{y}_j)}{p(\mathbf{y}_i) p(\mathbf{y}_j)} \right)$$
$$\propto p(\mathbf{f}^*)^{1-M} \prod_{i=1}^{M} p(\mathbf{f}^* | \mathbf{y}_i), \qquad (9)$$

corresponds exactly to the prediction of the Bayesian Committee Machine (BCM) [2]. This motivates the GPRF as a natural extension of the BCM as a model for the training set, providing an alternative to

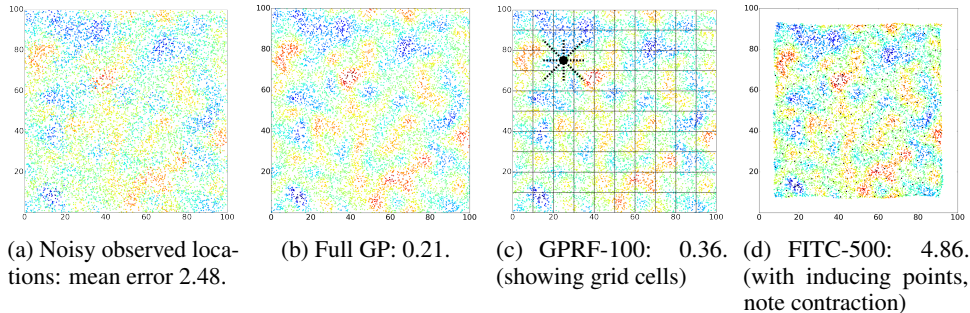

(a) Noisy observed locations: mean error 2.48.　　(b) Full GP: 0.21.　　(c) GPRF-100: 0.36. (showing grid cells)　　(d) FITC-500: 4.86. (with inducing points, note contraction)

Figure 2: Inferred locations on synthetic data ($n = 10000$), colored by the first output dimension $\mathbf{y}_1$.

the standard transductive interpretation of the BCM.[2] A similar derivation shows that the conditional distribution of any block $\mathbf{y}_i$ given all other blocks $\mathbf{y}_{j \neq i}$ also takes the form of a BCM prediction, suggesting the possibility of *pseudolikelihood* training [21], i.e., directly optimizing the quality of BCM predictions on held-out blocks (not explored in this paper).

## 4 Experiments

### 4.1 Uniform Input Distribution

We first consider a 2D synthetic dataset intended to simulate spatial location tasks such as WiFi-SLAM [22] or seismic event location (below), in which we observe high-dimensional measurements but have only noisy information regarding the locations at which those measurements were taken. We sample $n$ points uniformly from the square of side length $\sqrt{n}$ to generate the true inputs $X$, then sample 50-dimensional output $Y$ from independent GPs with SE kernel $k(r) = \exp(-(r/\ell)^2)$ for $\ell = 6.0$ and noise standard deviation $\sigma_n = 0.1$. The *observed* points $X^{\text{obs}} \sim N(X, \sigma_{\text{obs}}^2 I)$ arise by corrupting $X$ with isotropic Gaussian noise of standard deviation $\sigma_{\text{obs}} = 2$. The parameters $\ell$, $\sigma_n$, and $\sigma_{\text{obs}}$ were chosen to generate problems with interesting short-lengthscale structure for which GP-LVM optimization could nontrivially improve the initial noisy locations. Figure 2a shows a typical sample from this model.

For local GPs and GPRFs, we take the spatial partition to be a grid with $n/m$ cells, where $m$ is the desired number of points per cell. The GPRF edge set $E$ connects each cell to its eight neighbors (Figure 2c), yielding linear time complexity $O(nm^2)$. During optimization, a practical choice is necessary: do we use a fixed partition of the points, or re-assign points to cells as they cross spatial boundaries? The latter corresponds to a coherent (block-diagonal) spatial covariance function, but introduces discontinuities to the marginal likelihood. In our experiments the GPRF was not sensitive to this choice, but local GPs performed more reliably with fixed spatial boundaries (in spite of the discontinuities), so we used this approach for all experiments.

For comparison, we also evaluate the Sparse GP-LVM, implemented in GPy [23], which uses the FITC approximation to the marginal likelihood [7]. (We also considered the Bayesian GP-LVM [4], but found it to be more resource-intensive with no meaningful difference in results on this problem.) Here the approximation parameter $m$ is the number of inducing points.

We ran L-BFGS optimization to recover maximum a posteriori (MAP) locations, or local optima thereof. Figure 3a shows mean location error (Euclidean distance) for $n = 10000$ points; at this size it is tractable to compare directly to the full GP-LVM. The GPRF with a large block size ($m$=1111, corresponding to a 3x3 grid) nearly matches the solution quality of the full GP while requiring less time, while the local methods are quite fast to converge but become stuck at inferior optima. The FITC optimization exhibits an interesting pathology: it initially moves towards a good solution but then diverges towards what turns out to correspond to a contraction of the space (Figure 2d); we conjecture this is because there are not enough inducing points to faithfully represent the full GP

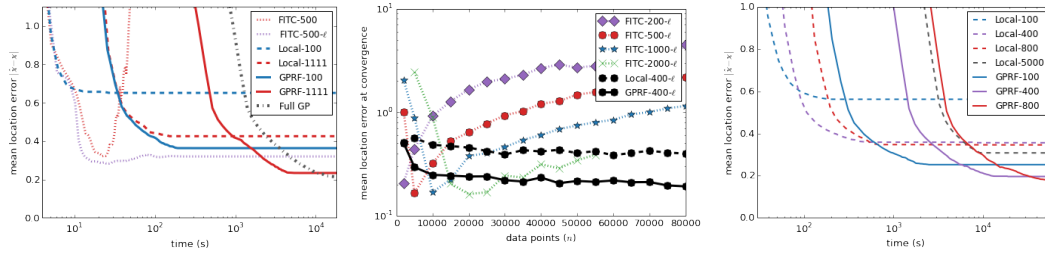

(a) Mean location error over time for $n = 10000$, including comparison to full GP.

(b) Mean error at convergence as a function of $n$, with learned lengthscale.

(c) Mean location error over time for $n = 80000$.

Figure 3: Results on synthetic data.

distribution over the entire space. A partial fix is to allow FITC to jointly optimize over locations and the correlation lengthscale $\ell$; this yielded a biased lengthscale estimate $\hat{\ell} \approx 7.6$ but more accurate locations (FITC-500-$\ell$ in Figure 3a).

To evaluate scaling behavior, we next considered problems of increasing size up to $n = 80000$.[3] Out of generosity to FITC we allowed each method to learn its own preferred lengthscale. Figure 3b reports the solution quality at convergence, showing that even with an adaptive lengthscale, FITC requires increasingly many inducing points to compete in large spatial domains. This is intractable for larger problems due to $O(m^3)$ scaling; indeed, attempts to run at $n > 55000$ with 2000 inducing points exceeded 32GB of available memory. Recently, more sophisticated inducing-point methods have claimed scalability to very large datasets [24, 25], but they do so with $m \leq 1000$; we expect that they would hit the same fundamental scaling constraints for problems that inherently require many inducing points.

On our largest synthetic problem, $n = 80000$, inducing-point approximations are intractable, as is the full GP-LVM. Local GPs converge more quickly than GPRFs of equal block size, but the GPRFs find higher-quality solutions (Figure 3c). After a short initial period, the best performance always belongs to a GPRF, and at the conclusion of 24 hours the best GPRF solution achieves mean error 42% lower than the best local solution (0.18 vs 0.31).

## 4.2 Seismic event location

We next consider an application to seismic event location, which formed the motivation for this work. Seismic waves can be viewed as high-dimensional vectors generated from an underlying three-dimension manifold, namely the Earth's crust. Nearby events tend to generate similar waveforms; we can model this spatial correlation as a Gaussian process. Prior information regarding the event locations is available from traditional travel-time-based location systems [26], which produce an independent Gaussian uncertainty ellipse for each event.

A full probability model of seismic waveforms, accounting for background noise and performing joint alignment of arrival times, is beyond the scope of this paper. To focus specifically on the ability to approximate GP-LVM inference, we used real event locations but generated synthetic waveforms by sampling from a 50-output GP using a Matérn kernel [3] with $\nu = 3/2$ and a lengthscale of 40km. We also generated observed location estimates $X^{\text{obs}}$, by corrupting the true locations with

$$\frac{\partial}{\partial \mathbf{x}_i} \log p(\mathbf{y}) = \frac{1}{2} \text{tr}\left( \left( (K_y^{-1}\mathbf{y})(K_y^{-1}\mathbf{y})^T - K_y^{-1} \right) \frac{\partial K_y}{\partial \mathbf{x}_i} \right)$$

involves the full precision matrix $K_y^{-1}$ which is not sparse in general. Bypassing this expression via automatic differentiation through the sparse Cholesky decomposition could perhaps allow exact GP-LVM inference to scale to somewhat larger problems.

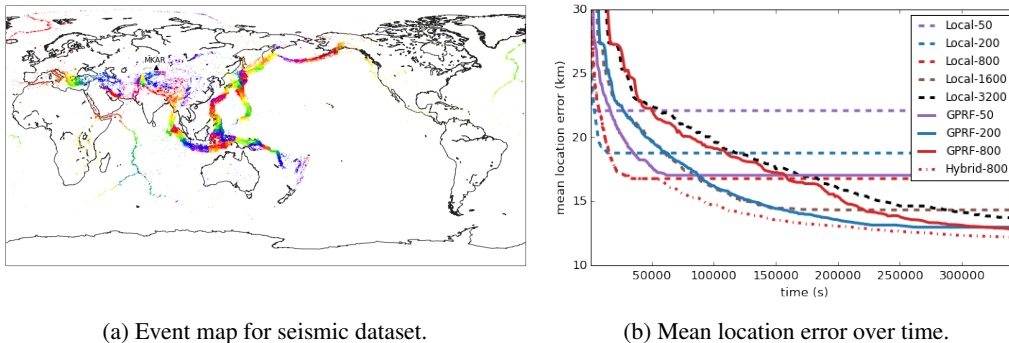

(a) Event map for seismic dataset.  (b) Mean location error over time.

Figure 4: Seismic event location task.

Gaussian noise of standard deviation 20km in each dimension. Given the observed waveforms and noisy locations, we are interested in recovering the latitude, longitude, and depth of each event.

Our dataset consists of 107556 events detected at the Mankachi array station in Kazakstan between 2004 and 2012. Figure 4a shows the event locations, colored to reflect a principle axis tree partition [27] into blocks of 400 points (tree construction time was negligible). The GPRF edge set contains all pairs of blocks for which any two points had initial locations within one kernel lengthscale (40km) of each other. We also evaluated longer-distance connections, but found that this relatively local edge set had the best performance/time tradeoffs: eliminating edges not only speeds up each optimization step, but in some cases actually yielded faster per-step convergence (perhaps because denser edge sets tended to create large cliques for which the pairwise GPRF objective is a poor approximation).

Figure 4b shows the quality of recovered locations as a function of computation time; we jointly optimized over event locations as well as two lengthscale parameters (surface distance and depth) and the noise variance $\sigma_n^2$. Local GPs perform quite well on this task, but the best GPRF achieves 7% lower mean error than the best local GP model (12.8km vs 13.7km, respectively) given equal time. An even better result can be obtained by using the results of a local GP optimization to initialize a GPRF. Using the same partition ($m = 800$) for both local GPs and the GPRF, this "hybrid" method gives the lowest final error (12.2km), and is dominant across a wide range of wall clock times, suggesting it as a promising practical approach for large GP-LVM optimizations.

## 5   Conclusions and Future Work

The Gaussian process random field is a tractable and effective surrogate for the GP marginal likelihood. It has the flavor of approximate inference methods such as loopy belief propagation, but can be analyzed precisely in terms of a deterministic approximation to the inverse covariance, and provides a new training-time interpretation of the Bayesian Committee Machine. It is easy to implement and can be straightforwardly parallelized.

One direction for future work involves finding partitions for which a GPRF performs well, e.g., partitions that induce a block near-tree structure. A perhaps related question is identifying when the GPRF objective defines a normalizable probability distribution (beyond the case of an exact tree structure) and under what circumstances it is a good approximation to the exact GP likelihood.

This evaluation in this paper focuses on spatial data; however, both local GPs and the BCM have been successfully applied to high-dimensional regression problems [11, 12], so exploring the effectiveness of the GPRF for dimensionality reduction tasks would also be interesting. Another useful avenue is to integrate the GPRF framework with other approximations: since the GPRF and inducing-point methods have complementary strengths – the GPRF is useful for modeling a function over a large space, while inducing points are useful when the density of available data in some region of the space exceeds what is necessary to represent the function – an integrated method might enable new applications for which neither approach alone would be sufficient.

### Acknowledgements

We thank the anonymous reviewers for their helpful suggestions. This work was supported by DTRA grant #HDTRA-11110026, and by computing resources donated by Microsoft Research under an Azure for Research grant.

## Footnotes

[1]The extension to multiple independent outputs is straightforward.

[2]The GPRF is still transductive, in the sense that adding a test block $\mathbf{f}^*$ will change the marginal distribution on the training observations $\mathbf{y}$, as can be seen explicitly in the precision matrix (7). The contribution of the GPRF is that it provides a reasonable model for the training-set likelihood even in the absence of test points.

[3]The astute reader will wonder how we generated synthetic data on problems that are clearly too large for an exact GP. For these synthetic problems as well as the seismic example below, the covariance matrix is relatively sparse, with only ~2% of entries corresponding to points within six kernel lengthscales of each other. By considering only these entries, we were able to draw samples using a sparse Cholesky factorization, although this required approximately 30GB of RAM. Unfortunately, this approach does not straightforwardly extend to GP-LVM inference under the exact GP, as the standard expression for the marginal likelihood derivatives

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
