[Supplementary Material]

# Gaussian Process Random Fields: Supplementary Material

**David A. Moore and Stuart J. Russell**

This file contains additional derivations for our NIPS 2015 paper, "Gaussian Process Random Fields". Notation used here follows the notation of the paper. Code to construct the datasets and reproduce the experimental results is available online at `https://github.com/davmre/gprf/`.

## 1 Block tree structure

It is straightforward to see that the GPRF objective is exact when the MRF induced by the true precision matrix $J$, with respect to our chosen partition of $\mathbf{y}$, is a tree. For any choice of root node $\mathbf{y}_{\text{root}}$, the tree structure implies that we can write the true GP distribution as a product of parent-conditional distributions,

$$p(\mathbf{y}) = p(\mathbf{y}_{\text{root}}) \prod_{i \neq \text{root}} p(\mathbf{y}_i | \mathbf{y}_{\pi(i)})$$

where $\pi(i)$ is the (unique) parent of node $i$ with respect to our chosen root. Then expanding the conditional distribution

$$p(\mathbf{y}) = p(\mathbf{y}_{\text{root}}) \prod_{i \neq \text{root}} \frac{p(\mathbf{y}_i, \mathbf{y}_{\pi(i)})}{p(\mathbf{y}_{\pi(i)})}$$

$$= p(\mathbf{y}_{\text{root}}) \prod_{i \neq \text{root}} p(\mathbf{y}_i) \frac{p(\mathbf{y}_i, \mathbf{y}_{\pi(i)})}{p(\mathbf{y}_i) p(\mathbf{y}_{\pi(i)})}$$

$$= \left( \prod_i p(\mathbf{y}_i) \right) \left( \prod_{i \neq \text{root}} p(\mathbf{y}_i) \frac{p(\mathbf{y}_i, \mathbf{y}_{\pi(i)})}{p(\mathbf{y}_i) p(\mathbf{y}_{\pi(i)})} \right)$$

yields exactly the GPRF objective for the edge set $E = (i, \pi(i))$, i.e., the edges that define the tree.

Note that the structure of the MRF induced by the true GP will depend on the partition we choose: a given precision matrix may induce a tree structure for some choices of partition but not for others (e.g., even a fully dense matrix can be viewed as a tree for trivial partitions that split the dataset into only one or two blocks).

In many cases it is easier to reason about the structure of the covariance matrix than that of the precision matrix. Assuming a stationary kernel, nonzero (or non-negligible) entries of the covariance matrix correspond to data points that are nearby to each other, meaning that the sparsity pattern of the covariance matrix reflects the geometry of the data itself. If the data can be viewed as lying on a treelike manifold – for example, seismic fault lines, or even trivial special cases such as time series data which lies on the real line – then for reasonable choices of partition, a graph connecting nearby blocks of data points will have a tree structure. Of course, there is no formal guarantee that this structure will fully carry over to the precision matrix, though intuitively we'd expect that points very distant from each other are also unlikely to interact strongly in the precision matrix.

## 2 Approximation to the true Gaussian

In this section we prove Theorem 1 from the main text, showing that $q_{GPRF}$ is an unnormalized Gaussian density with a particular precision matrix.

For any pair of blocks $(i,j)$, define the *local precision matrix* $Q^{(ij)}$ to be the inverse of the marginal covariance,

$$Q^{(ij)} = \begin{pmatrix} Q_{11}^{(ij)} & Q_{12}^{(ij)} \\ Q_{21}^{(ij)} & Q_{22}^{(ij)} \end{pmatrix} = \begin{pmatrix} K_{ii} & K_{ij} \\ K_{ji} & K_{jj} \end{pmatrix}^{-1},$$

The notation $Q^{(ij)}$ is used to distinguish these local precision matrices from the blocks $J_{ij}$ of the global precision matrix. Writing $q_{GPRF}$ in terms of unnormalized Gaussian densities,

$$\log q_{GPRF}(\mathbf{y}) = -\frac{1}{2}\sum_{i=1}^{M}(1-|E_i|)\mathbf{y}_i^T K_{ii}^{-1}\mathbf{y}_i - \frac{1}{2}\sum_{(i,j)\in E}\begin{pmatrix}\mathbf{y}_i\\\mathbf{y}_j\end{pmatrix}^T Q^{(ij)}\begin{pmatrix}\mathbf{y}_i\\\mathbf{y}_j\end{pmatrix} + C$$

$$= -\frac{1}{2}\sum_{i=1}^{M}\mathbf{y}_i^T\left(K_{ii}^{-1} - |E_i|K_{ii}^{-1}\right)\mathbf{y}_i - \frac{1}{2}\left(\sum_{(i,j)\in E}\mathbf{y}_i^T Q_{11}^{(ij)}\mathbf{y}_i + 2\mathbf{y}_i^T Q_{12}^{(ij)}\mathbf{y}_j + \mathbf{y}_j^T Q_{22}^{(ij)}\mathbf{y}_j\right) + C$$

$$= -\frac{1}{2}\sum_{i=1}^{M}\mathbf{y}_i^T\left(K_{ii}^{-1} + \sum_{j\in E_i}\left(Q_{11}^{(ij)} - K_{ii}^{-1}\right)\right)\mathbf{y}_i - \sum_{(i,j)\in E}\mathbf{y}_i^T Q_{12}^{(ij)}\mathbf{y}_j + C$$

we obtain the standard form of a Gaussian MRF (expression (3) from the main text) showing that $q_{GPRF}$ does in fact induce a Gaussian density on $\mathbf{y}$. Note that in passing from the second to the third line we used the fact that $Q_{11}^{ij} = Q_{22}^{ji}$, by definition. This Gaussian representation allows us to read off the implicit precision matrix $\tilde{J}$ in block wise form

$$\tilde{J}_{ii} = K_{ii}^{-1} + \sum_{j\in E_i}\left(Q_{11}^{(ij)} - K_{ii}^{-1}\right), \qquad \tilde{J}_{ij} = \begin{cases} Q_{12}^{(ij)} & \text{if } (i,j)\in E \\ 0 & \text{otherwise.} \end{cases} \tag{1}$$

We see that the off-diagonal blocks of the precision matrix are simply the corresponding blocks of the pairwise local precisions. Each diagonal block, by contrast, combines the inverse of the local covariance matrix with corrections from the pairwise precisions. Note that the approximate precision $\tilde{J}$ may not be positive definite. In this case $q_{GPRF}$ is not a normalizable density, although it is still "approximately normalized" in the sense of the next section.

## 3  Approximate normalization

In this section we prove Theorem 2 from the main text:

**Theorem 2.** *The objective $q_{GPRF}$ is approximately normalized in the sense that the optimal value of the* Bethe free energy *[1],*

$$F_B(b) = \sum_{i\in V}\left(\int_{\mathbf{y}_i} b_i(\mathbf{y}_i)\frac{(1-|E_i|)\ln b(\mathbf{y}_i)}{\ln\psi_i(\mathbf{y}_i)}\right) + \sum_{(i,j)\in E}\left(\int_{\mathbf{y}_i,\mathbf{y}_j} b_{ij}(\mathbf{y}_i,\mathbf{y}_j)\ln\frac{b_{ij}(\mathbf{y}_i,\mathbf{y}_j)}{\psi_{ij}(\mathbf{y}_i,\mathbf{y}_j))}\right) \approx \log Z,$$

$$\tag{2}$$

*the approximation to the normalizing constant found by loopy belief propagation, is precisely zero. Furthermore, this optimum is obtained when the pseudomarginals $b_i, b_{ij}$ are taken to be the true GP marginals $p_i, p_{ij}$.*

*Proof.* These claims are established rather directly by substituting the GPRF potentials $\psi_i^{GPRF}, \psi_{ij}^{GPRF}$ for the log pseudomarginals $\log\psi_i, \log\psi_{ij}$ in (2), yielding

$$F_B(b) = \sum_{i\in V}\left(\int_{\mathbf{y}_i} b_i(\mathbf{y}_i)\frac{(1-|E_i|)\ln b_i(\mathbf{y}_i)}{(1-|E_i|)\ln p(\mathbf{y}_i)}\right) + \sum_{(i,j)\in E}\left(\int_{\mathbf{y}_i,\mathbf{y}_j} b_{ij}(\mathbf{y}_i,\mathbf{y}_j)\ln\frac{b_{ij}(\mathbf{y}_i,\mathbf{y}_j)}{p(\mathbf{y}_i,\mathbf{y}_j))}\right)$$

$$= \sum_{i\in V}KL[b_i\|p_i] + \sum_{(i,j)\in E}KL[b_{ij}\|p_{ij}], \tag{3}$$

where $KL[b\|p] = \int b(\mathbf{x})\ln\frac{b(\mathbf{x})}{p(\mathbf{x})}d\mathbf{x}$ is the Kullback-Liebler divergence between distributions $b$ and $p$. This is minimized when the distributions are equal, at which point the divergence is zero. Thus, taking $b_i = p_i$ and $b_{ij} = p_{ij}$ yields the optimal value $F_B = 0$. □

We might have hoped that, as local GPs match the marginal distributions of the full GP on individual blocks, perhaps a higher-order approximation could match the exact marginals on pairs of blocks. This is not possible, since any Gaussian distribution whose pairwise marginals match the full GP must in fact be the full GP (Gaussians are entirely characterized by their covariances). Instead we can view $q_{GPRF}$ as *approximately* matching the pairwise marginals of the full GP, in the sense that the pseudomarginals found by running loopy belief propagation on $q_{GPRF}$ are in fact the true marginals of the full GP. This is a consequence of the fact that loopy BP converges to stationary points of the Bethe energy [1].

## References

[1] Jonathan S Yedidia, William T Freeman, and Yair Weiss. Bethe free energy, Kikuchi approximations, and belief propagation algorithms. *Advances in Neural Information Processing Systems (NIPS)*, 13, 2001.