[Reviews · NeurIPS 2015]

Submitted by Assigned_Reviewer_1

*Update after the author's rebuttal*

The results reported in the rebuttal are very interesting and I very much hope the authors include the comparisons and analyses mentioned there in the final version of the paper (if it gets accepted).

=== 0. Summary

This paper presents an approximation method to Gaussian processes models (mainly focused on the Gaussian process latent variable model

GPLVM), based on the "partitioning" approach, where local GPs are fitted to

subsets of the data. The main claimed contribution is that of an "efficient" marginal likelihood approximation based on linking up the local GPs through a Gaussian MRF.

1. Clarity The paper is well written with a clearly stated contribution, that of providing a new approximation for GP inference.

2. Quality, originality and significance The paper seems technically sound as the objective function approximating the marginal likelihood, although unnormalized, has some interesting properties: (a) that of recovering the exact marginal likelihood for a tree-structure over the partitions, and (b) that of approximating the marginals p(y_i) and p(y_i, y_j). There are, however, two major deficiencies of this paper: (i) there is some important missing related work and (ii) it should have compared to a better baseline model.

Regarding the first deficiency, scaling up GPs, there have been important advances in the last couple of years or so, most notably the work in [1], and subsequent work by other authors. The work in [2] does provide a probabilistic framework for "partitioning" models, while also learning the data-point assignment within a variational framework. Finally, the work in [3] not only provides a distributed approach for large-scale inference but also addresses GPLVM problems. The authors should not only acknowledge these contributions but also carry out comparisons with at least [3]. This brings up the second deficiency, not comparing to a suitable baseline model. Even if [3] is not used, it seems that the authors compare to vanilla GPLVM. It has been shown that the Bayesian GPLVM of Titsias et al (reference 4 in the paper) is a better model and has the same computational cost as the simpler sparse version of Lawrence (reference 7 in the paper). Therefore, it looks like the authors are using a very weak baseline.

It is interesting that the proposed model can be seen as an extension of the BCM. However, it inherits its limitation as a transductive method.

Another drawback of the paper is its computational cost, which prohibits its applicability to truly large scale applications (in contrast to the references below [1], [2], and [3]). Indeed, n^2m seems too costly to be considered a practical algorithm for large scale problems, and the size of the datasets of the experiments corroborate that. I do encourage the authors to explore the avenue of scaling further their approach.

With regards to the experiments, it will be good to see some results regarding hyper-parameter learning and see how the proposed approach compares to the state of the art.

3. Minor comments - sentence on lines 106-107 does not make sense on its own: "by minimizing the KL divergence to the full GP" - What happened to Q_{22}^{(ij)} when going from the Equation in line 234 to the equations below (including Equation 7)?

4. References

[1] James Hensman, Nicolo Fusi, Neil D. Lawrence. in UAI 2013. Gaussian Processes for Big Data. in UAI 2013.

[2] T. V. Nguyen and E. V. Bonilla. Fast Allocation of Gaussian Process Experts. in ICML 2014.

[3] Yarin Gal, Mark van der Wilk, and Carl Rasmussen. Distributed variational inference in sparse Gaussian

process regression and latent variable models. In NIPS 2014.
Summary: A method for approximating the marginal likelihood in GP models by linking up local GPs with a Gaussian MRF. The

objective function has interesting properties but the authors fail to cite some important related work and to compare to more reasonable baselines. The computational cost of the method is too high to be considered practical for large scale applications.

Submitted by Assigned_Reviewer_2

To apply Gaussian process to large-scale data is no doubt a hot topic in recent years. The author(s) is encouraged to provide a more complete survey on related works.

The proposed idea is simple and intuitive, which aims to link GPs built for each data partition blocks together by a random field. A key point to address (or to clarify) is to explain when and how the precision matrix induces a tree-structure (or close to that) on the partition blocks; or whether or not the model can still work even it is not close to a tree-strucure. (line 196 to line 200) For real data, it is interesting to see how the data may obey or violate the tree-structure assumption, or if the data violate the assumption, can loopy BP work? My opinion is to explore some spatial or temporal relationships in the data and to explain how the random field can fit better to real data given the spatial or temporal relationships. The point is important to because it decides how the proposed method may work or may not work for a wide range of applications.

It will be good if the authors can compare the proposed GPRF to Zhong et al. [17] in more details.
Summary: The work is about a Gaussian process approximation given large-scale data. The idea is simple. What needs to be clarified is when and how the precision matrix induces a tree-structure (or close to that) on the partition blocks

Submitted by Assigned_Reviewer_3

This paper proposes a new approximate scalable inference algorithm for both supervised and unsupervised GP models. It introduces pairwise potentials on top of the local GP models and make the entire model of output variables a pairwise Gaussian MRF. The pairwise potential couples different local GPs and is useful to provide a smooth predictive distribution. The main approximation introduced by the author is to replace the exact marginal likelihood with a Bethe-type objective function which can be computed in O(n^2 m) time, faster than the full GP, and shows that the predictive distribution for a test point is equivalent to BCM. This is quite a novel and interesting idea. I think it might trigger further study on how to borrow other approximate inference algorithms for regular MRFs to the GP regression problem.

The main shortcoming of this method in my opinion is the quadratic time complexity on n when one uses a fully connected graph as used in the experiments. The typical time complexity for most sparse GP algorithms is O(n m^2) and still that is not suitable for real large scale problems nowadays. Therefore, I don't think the new algorithm with the O(n^2 m) complexity could scale well for large applications. Also, the authors do not discuss how to propose a sparse structure in the GRF.

I have another concern about approximate objective function q. Is the covariance matrix of q possible to be not positive definite? If so, it would be useful to the authors to discuss the potential problems caused by a non-PD covariance matrix. Also, the notion of approximate normalization is hard to quantify as the Bethe approximation in loopy BP could be quite bad depending on the strength of loops. As a minor comment, it is a little surprising that the q distribution is approximation to the full GP but the pseudo-marginals obtained from loopy BP actually matches the exact distribution.

In the experiment, GPRF shows a better accuracy than local GP models. But it is not really a fair comparison for two models with the same number of group size because the time complexity of GPRF is much higher than local GP. A better comparison should be two models with different group size but similar time complexity. Also, as GPRF is proposed as a scalable GP inference algorithm, it is necessary to compare with standard sparse GP algorithms such as GP-LVM with FITC approximation or the sparse variational inference algorithm.
Summary: This paper considers a different approach to improving the scalability of GP inference from standard algorithms with a Bethe approximation. While the time complexity of the current algorithm with a fully connect graph is still higher than regular local GP and sparse GP approximate algorithms, the idea of using approximate inference algorithms from the MRF area is quite novel and worth further investigation.

Submitted by Assigned_Reviewer_4

This paper proposes a GPRF model using pairwise potentials for approximate alternative of large-scale GP. The idea is interesting, yet there are several key problems make the claims not solid enough. The main problems in the reviewer's opinion are listed as below: 1. There lacks enough survey and comparison to other multiple GPs baselines besides "local GPs". For example in literature, the blockwise or sparse assumption in a full GP and the mixture of GP experts both have the dependency between different GPs. Since the proposed GPRF is under the same motivation, it is natural and fair to compare with them besides independent local GPs. 2. There are several key featured settings in GPRF that tend to affect the performance seriously, while not mentioned in details. For instance, the pairwise edge set E determines the complexity and approximation accuracy, and it is not mentioned what kind of E is used in experiments. As another example, the number of partitions L also affects the results. However, in the plotted figures, why sometimes a larger L leads to a better result, while sometimes it is reversed? If there is not a monotonic relation, it is expected to analyze how many should L be in what scenarios. Moreover, if the hyperparameters are important in resulted performance, how to choose them appropriately, and how do they affect the results? 3. Another drawback is the computational complexity, which cannot scale well on real applications.
Summary: This paper proposes a GPRF model using pairwise potentials for approximate alternative of large-scale GP. The idea is interesting, yet there are several key problems make the claims not solid enough.

Author Feedback
Author rebuttal: Thanks to all the reviewers for your thoughtful feedback!

Several reviewers expressed concerns about O(n^2m) scaling. As mentioned (perhaps too fleetingly) in the paper, using a spatially-local edge set reduces the cost to O(nm^2), i.e., linear in n, the same as sparse methods (though with different scaling assumptions, see below). This is in fact what we did for all the experiments, which will certainly be clarified (in particular, we find partitions using RPC, then draw an edge between any two blocks containing initial locations less than two lengthscales apart). For problems with short-lengthscale structure, almost all relevant interaction is between nearby blocks, so very little is lost by reducing the edge set in this way. Post submission we have been able to run synthetic experiments up to 80000 points, in four hours on a single machine. At this size (and for 30k, 40k, 60k points) the locations recovered by the best GPRF optimizations are 5-10% better in mean average deviation than those of the best local GP models given the same time constraint.

Reviewers also encouraged comparisons to other methods, in particular sparse/inducing point approximations and mixture-of-experts models. We are not aware of mixture-of-experts being applied to the GPLVM setting, but note that for tractability reasons, mixture-of-experts models often use a MAP assignment of points to experts (e.g., Nguyen and Bonilla, cited by reviewer 1) which makes them in practice an intelligent way of choosing the clusters for a local GP model. In that sense they are somewhat orthogonal to though obviously related to our work.

We have been able to compare directly to inducing point methods, the Sparse/FITC and Bayesian GPLVMs as implemented in GPy (http://sheffieldml.github.io/GPy/). The sparse GPLVM outperformed both local GPs and the GPRF on our smaller synthetic problems (up to n=15k), but it required a large number of inducing points (m=1000 for the 15k dataset) to do so. For the 20k and larger datasets, the sparse GPLVM required an intractable number of of inducing points to form an accurate representation; the best results we achieved within a four-hour time bound were ~10x worse than local and GPRF models. The Bayesian GPLVM performed similarly to the sparse GPLVM given the same inducing point capacity, but ran out of memory for m > 400, so it was not competitive on large datasets. Recent work (e.g. Gal et al. cited by reviewer 1) scales these methods to very large n, but with comparatively small m (up to roughly 200); since the optimization over inducing points is not distributed we would expect the same blowup in runtime as for the simpler sparse methods. These results are consistent with the claim in the paper that inducing point methods are not suited to modeling short-lengthscale structure; they function well in scaling regimes where the world is "growing denser", so that much of the data can ultimately be thrown away, while local methods and the GPRF scale well when the world is "growing larger" and we need to represent an increasing volume of space while maintaining local detail. We are interested in the latter regime.

Some other specific points:

Reviewer 1, fate of Q_{22}^{(ij)} after line 234: it is implicitly rewritten as Q_{11}^{(ji)}, which is equivalent. Intuitively the sum over edges in line 234 counts every edge once, while the sum over neighbors in line 238 counts every edge twice (if A is a neighbor of B, then B is also a neighbor of A), so the expression itself needs only represent one "side" of the edge.

Reviewer 2, non-PD covariance matrices: it is indeed possible for the implicit precision matrix to be non-PD. In this case the objective is still "approximately normalized" in the sense of the Bethe energy but is not actually normalizeable. We have not studied this phenomenon in detail, but it does seem interesting to understand! (and certainly worth mentioning in the paper).

Reviewer 4, comparison to Zhong et al: their work is not directly comparable; we mentioned it only due to the similar name. They use an MRF as a regularizing prior on the latent points; the likelihood itself is just a standard GPLVM.

Reviewer 5, relation to GMRFs: the MRFs we consider are Gaussian, so they are GMRFs. In this sense the GPRF is a recipe for constructing a particular (unnormalized) GMRF to approximate the marginal likelihood of a GP latent variable model.

Reviewer 8, number of partitions: there is a statistical/computational tradeoff; larger block sizes yield better solutions at their optima but given finite time the optimizer will take fewer steps and may not reach the optimum. The same tradeoff exists in local GPs, and an analogous one in choosing the inducing point parameter for sparse methods.